# CROS or hearing aid? Selecting the ideal solution for unilateral CI patients with limited aidable hearing in the contralateral ear

Sarah Lively[1]*, Smita Agrawal[2], Matthew Stewart[1¤a], Robert T. Dwyer[2], Laura Strobel[1¤b], Paula Marcinkevich[1], Chris Hetlinger[3], Julia Croce[1]

1 Department of Otolaryngology, Thomas Jefferson University Hospital, Philadelphia, PA, United States of America, 2 Collaborative Research Group, Clinical Research, Advanced Bionics, Valencia, CA, United States of America, 3 Research and Technology Group, Advanced Bionics, Valencia, CA, United States of America

¤a Current address: Internal Medicine, Cleveland Clinic Foundation, Cleveland, OH, United States of America
¤b Current address: Audiology Associates Hawaii, Aiea, Hawaii, United States of America
* sarah.lively@jefferson.edu

**Data Availability Statement:** All relevant data are within the paper and its Supporting Information files.

## Abstract

A hearing aid or a contralateral routing of signal device are options for unilateral cochlear implant listeners with limited hearing in the unimplanted ear; however, it is uncertain which device provides greater benefit beyond unilateral listening alone. Eighteen unilateral cochlear implant listeners participated in this prospective, within-participants, repeated measures study. Participants were tested with the cochlear implant alone, cochlear implant + hearing aid, and cochlear implant + contralateral routing of signal device configurations with a one-month take-home period between each in-person visit. Audiograms, speech perception in noise, and lateralization were evaluated. Subjective feedback was obtained via questionnaires. Marked improvement in speech in noise and non-implanted ear lateralization accuracy were observed with the addition of a contralateral hearing aid. There were no significant differences in speech recognition between listening configurations. However, the chronic device use questionnaires and the final device selection showed a clear preference for the hearing aid in spatial awareness and communication domains. Individuals with limited hearing in their unimplanted ears demonstrate significant improvement with the addition of a contralateral device. Subjective questionnaires somewhat contrast with clinic-based outcome measures, highlighting the delicate decision-making process involved in clinically advising one device or another to maximize communication benefits.

## Introduction

Unilateral cochlear implantation (CI) is the standard of care for adults with severe, profound, or moderate sloping to profound bilateral sensorineural hearing loss (SNHL) [1]. A cochlear implant (CI), widely recognized as the most successful neural prosthesis, bypasses an individual's poor cochlear function by using current to electrically stimulate the remaining auditory

**Funding:** This investigator-initiated study was supported by Advanced Bionics via a research grant and an equipment loan for study conduction. Participants had the choice to keep their preferred study device (a hearing aid or CROS) at the end of the study. The specific roles of the co-authors are articulated in the 'author contributions' section of this manuscript.

**Competing interests:** I have read the journal's policy and the authors of this manuscript have the following competing interests: RD, CH, and SA are employees of Advanced Bionics. The remaining authors declare no conflicts of interest. This does not alter our adherence to PLOS ONE policies on sharing data and materials.

fibers, creating the perception of sound. In the United States alone, there are 170,252 individuals with at least one CI [2]. According to the voluntary reports of registered cochlear implant devices to the U.S. Food and Drug Administration, approximately 736,900 devices have been implanted worldwide as of December 2019 [3].

While outcomes with a single CI are impressive, a large body of evidence shows the benefits of bilateral cochlear implants (BiCI). The benefits from the addition of a second CI arise from binaural summation effects [4–6], access to interaural level difference (ILD) cues resulting in improved spatial hearing abilities [7–9], and enables patients to benefit from the head shadow no matter from which side speech originates. For example, the head shadow effect puts unilateral listeners at a disadvantage when undesired signals are presented to the listener's implanted ear [10] or when the signal of interest (usually speech) is presented to the unimplanted ear [11]. Despite the benefits of bilateral cochlear implantation, a second cochlear implant (CI) surgery is often not an option for many reasons, including insurance coverage or medical contraindications [12]. Consequently, there is a large population of unilateral adult cochlear implant users.

For individuals who cannot obtain a second CI, utilizing a HA or a contralateral routing of sound (CROS) device remain viable options. While the degree of benefit can depend on the severity of hearing loss in the non-implanted ear, bimodal listening can significantly benefit unilateral CI listeners with little aidable hearing in the contralateral ear [13, 14]. Documented benefits include benefits in speech understanding, localization, sound quality, ease of hearing, and music enjoyment [15, 16]. Alternatively, research has shown the benefits of CROS devices [12, 17–21]. In a CROS system, sound is routed from a device on the poorer ear and transmitted to a device worn on the contralateral ear. For individuals with no aidable hearing in the contralateral ear, a CROS device can help overcome the limitations of the head shadow.

From a clinical perspective, rehabilitating unilaterally implanted individuals is not straightforward. In the present study, the population of interest was bimodal listeners with limited aidable hearing in the non-implanted ear. These listeners may benefit from bilateral device use, but it may not be apparent from their pure-tone audiogram if they will benefit from bimodal listening or if a CROS device would provide better outcomes. Holder et al. [22] recommended first fitting these patients with a HA before trial with CROS, however, currently, there are no guidelines available to make a more informed decision based on data that could be obtained during a clinical visit. In the present study, we collected (1) non-CI ear unaided pure-tone audiogram, (2) speech perception in noise with CI alone, CI+CROS, and CI+HA, (3) lateralization of sound sources with CI alone, CI+CROS, and CI+HA, (4) subjective questionnaires after take home phases with each device, and (5) participant-reported device preference at the end of the study to determine which device may provide more benefit beyond unilateral CI listening alone. We hypothesize that a test battery of clinic-based outcome measures (e.g., speech tests, lateralization, etc.) and subjective questionnaires may be needed to assist in selecting the appropriate device.

## Materials and methods

### Participants

The present study employed a prospective, within-participants, repeated measures design. All procedures were reviewed and approved by the Thomas Jefferson University Institutional Review Board (IRB) before participant recruitment (IRB approval #: 19P.195). Eighteen adult CI recipients were recruited from the Jefferson Balance and Hearing Center and enrolled in the study after obtaining informed written consent between September 2019 and October 2019. As the authors obtained informed consent and collected data for this prospective study,

**Table 1. Participant demographics.**

| Study ID | Age (yrs.) | Implant Type/ electrode | Duration of CI experience (yrs; mos.) | Pre-study contralateral ear device | Duration of current HA or CROS use | Strategy |
|---|---|---|---|---|---|---|
| S01 | 51 | Ultra / Mid-Scala | 3; 6 | Phonak Audeo V90 RIC | 2; 9 | S |
| S02 | 79 | Ultra / Mid-Scala | 1; 9 | Naída Link RIC | 1; 9 | S |
| S03 | 84 | Ultra / Mid-Scala | 1; 6 | Starkey BTE Z-series | 2; 0 | S |
| S04 | 79 | 90K Advantage / Mid-Scala | 5; 9 | Phonak Naída Link RIC | 4; 2 | S |
| S05 | 51 | 90K / Mid-Scala | 3; 5 | Phonak Naída Link CROS | 2; 0 | S |
| S06 | 60 | 90K / 1J | 8; 9 | Phonak Naída Link RIC | 1; 9 | S |
| S07 | 74 | 90K Advantage / Mid-Scala | 4; 3 | CI only | 0 | S |
| S08 | 73 | Ultra / Mid-Scala | 1; 2 | Naída Q50 UP BTE | 2; 6 | S |
| S09 | 66 | Ultra / Mid-Scala | 2; 8 | Naída Link RIC | 2; 0 | S |
| S12 | 81 | 90K Advantage / Mid-Scala | 3; 0 | Naída Link UP | 2; 0 | S |
| S13 | 78 | CII / 1J | 16; 10 | Naída Link UP | 1; 9 | S |
| S14 | 73 | 90K Advantage / Mid-Scala | 1; 6 | Naída Link RIC | 0; 9 | S |
| S16 | 66 | Ultra 3D / Mid-Scala | 1; 0 | Naída Link RIC | 0; 9 | S |
| S17 | 81 | Ultra / Mid-Scala | 1; 9 | Naída Link UP | 1; 3 | S |
| S18 | 76 | Ultra 3D / Mid-Scala | 1; 5 | Starkey Z series RIC | 1; 5 | S |
| S19 | 81 | 90K / 1J | 7; 9 | Naída Link RIC | 1; 0 | P |
| S20 | 49 | Ultra 3D / Mid-Scala | 16; 10 | Naída Link UP | 1; 7 | P |
| S21 | 77 | 90K / 1J | 9; 1 | Naída Link CROS | 2; 0 | P |
| **Mean** | 71 | - | | | | 15 S, 3 P |

MS = Mid-Scala, RIC = receiver in canal, BTE = behind the ear, UP = UltraPower, S = sequential, P = paired

they had access to information that could identify individual participants during or after data collection. The study protocol ensured the privacy and confidentiality of the participants' information.

Qualifying participants met the following candidacy criteria: adults ($>$ 18-years of age), unilaterally implanted with an Advanced Bionics (Valencia, CA, USA) CI (CII internal or newer), at least six months of CI experience, non-CI ear audiometric thresholds $\leq$ 100 dB HL at and below 500 Hz, CI only speech perception score in quiet with Institute of Electrical and Electronics Engineers (IEEE) sentences [23] $\geq$ 40% and ability and willingness to participate in multiple sets and sessions of open-set speech testing. Current use of a HA or CROS device in the non-implanted ear was not a disqualifier.

Participant demographics are shown in Table 1. At the time of enrollment, the mean age of the participants was 71 years (range 49 to 84 years). Eight participants were male, and ten were female. All participants used a Naída[TM] CI Q90 sound processor except one participant (S_19), who used a Naída CI Q70.

## Testing schedule

Between the three office visits were two "take-home" phases of four weeks each. The intent of the take home periods was an attempt to control for auditory experience as a confounding factor in the comparison of two listening configurations. At visit one, each participant completed baseline assessments with a study-issued sound processor in the CI alone listening

configuration and was then assigned a study-issued HA or CROS study device. After the initial 4-week take-home phase, participants completed a performance assessment with their assigned device. They were then fit with the alternate device that they then used during the second take-home period. During visit 3, participants completed a second assessment of outcomes with their second device. Final subjective assessments were also conducted at this time. For the purposes of this study, 'benefit' is defined as the difference in outcomes achieved with the CI alone listening configuration compared to listening configurations that employ the use of a contralateral device.

## Study devices and programming

Participants completed all sound booth testing and take-home phases with study-issued devices (i.e., AB Naída CI Q90 sound processor, Phonak Naída Link CROS, Phonak Naída Link UP BTE) containing the settings from their preferred listening program, which had been established over the course of routine clinical appointments with her or his audiologist. All participants used the current-steering based HiRes[TM] Optima sound coding strategy [24] (15 HiRes sequential, 3 HiRes paired). All participants used ClearVoice[TM] [25], a speech enhancement strategy, at medium strength and microphone mode set to T-Mic only in their everyday program. No modifications to electrical stimulation parameters, sound-coding strategy, microphone source, or ClearVoice strength were modified during this study.

The CROS system employs Phonak's proprietary Hearing Instrument Body Area Network wireless technology (HIBAN); when used in conjunction with the CI, the signal from both devices is combined in a 50/50 mix, and an adjustment to the T-Mic is applied. This adjustment results in a gain of 1–2 dB from 1–3 kHz, 5 dB gain from 3 to 5 kHz, and up to 8 dB gain at 7 kHz relative to the T-Mic input without a CROS device [26]. While the CROS device did not require individualized fitting, the Naída Link HA required Phonak Target Software v6.1 to fit the HA to the participant's hearing loss. Adaptive Phonak Digital Bimodal (APDB) fitting was verified using NAL-NL2 targets at 65 dB SPL speech input [27]. Soundflow (a classifier-based, automatic program) was not active in the HA program, and the microphone mode used was Real Ear Sound, which aims to restore the natural directivity pattern of the outer ear by selectively applying directionality to high frequencies.

## Test measures

**Pure-tone audiometry.** During visit 1, pulsed pure tones from 125 Hz to 8000 Hz (including inter-octaves) were presented via Etymotic Research ER3A insert earphones (Elk Grove Village, IL, USA) to obtain unaided thresholds in the non-implanted ear. Bone conduction thresholds were acquired from 500 Hz to 4000 Hz at octave frequencies.

**Speech perception.** To determine study candidacy, speech perception was measured in quiet using two lists of 10 IEEE sentences spoken by a male talker presented at 60 dBA from a loudspeaker located 1 meter in front of the listener. Speech materials were delivered via a custom research software (LIST Player Ver. 3, Advanced Bionics, Valencia, CA, USA) and a Madsen Astera[2] clinical audiometer (Middleton, WI, USA).

Speech perception in noise testing was conducted to assess the benefit of each hearing configuration (CI only vs. CI+HA vs. CI+CROS) using IEEE sentences presented at 60 dbA as the target stimuli. The noise stimuli was comprised of a 2-talker babble for noise created by extracting and combining AzBio sentences [28] spoken by two female talkers. Participants were positioned in a sound booth so that the two wall-mounted loudspeakers were at ± 60˚ and 1-meter from the participant. Target stimuli were presented from the speaker directed towards the non-CI ear and noise from the speaker towards the CI ear. Individualized noise

presentation levels were determined for each participant. They were identified as the noise level which the listener's CI only score (in percent correct) with IEEE sentences was about 50% of their score in quiet. This individualized signal-to-noise ratio (SNR) was used for all subsequent speech-in-noise testing during all in office visits.

**Lateralization.** Sound lateralization ability was assessed in each of the three hearing configurations using a 3-second duration pink noise stimulus presented randomly from either the left or right speaker positioned at ± 60˚. Prior to data collection, participants were acclimated to the task. Twenty trials were presented via ListPlayer at 60 dBA with +/5 dB across trial level roving. Participants were instructed to verbally identify whether they perceived the stimulus to be originating from the left or right speaker. Results are reported in percent correct.

**Questionnaires.** Participants completed the "Perceived Bimodal Benefit" and the "Baseline Needs Assessment" questionnaires during visit one. These custom questionnaires assessed subjective outcomes regarding commonly encountered hearing situations in quiet and noise, sound quality, sound awareness, and listening effort. Custom questionnaires designed to address specific characteristics of an intervention have been documented in the literature [29, 30]. Prior to this study, our questionnaires were developed with assistance from audiologists and other experts familiar with the literature to ensure that the questions were relevant and appropriate before being piloted in a small group of individuals to identify any issues with the questions and ensure that they are clear and understandable. To ensure the reliability of the data, all questionnaires were administered in a consistent manner to all participants. Ratings were made on a 7-point Likert scale ranging from "1-extremely disagree" to "7-extremely agree." Additionally, after each take-home period, participants completed a chronic device use questionnaire for each device. These questionnaires are included in the S1 File.

**End of study device preference.** Participants indicated the preferred hearing device configuration at the end of the study. After they indicated their preference and completed study participation, participants had the option to keep their preferred device (HA or CROS).

**Statistical analysis.** Statistical analyses were completed using SigmaPlot (version 12.5, Inpixon, Palo Alto, CA, USA). We used an alpha level of .05 for all statistical tests. Data were analyzed using repeated-measures ANOVA and paired t-tests with Bonferroni correction to reduce the chance of type 1 error. Unless otherwise noted, the mean (standard deviation) of results are reported. Linear mixed effect models to analyze the effect of starting group (CROS or HA) on the results were completed using R (version 4.1.3). Linear mixed-effects models were fitted using the 'lme4' package (version 1.1–32) for modeling fixed and random effects [31]. The 'lmerTest' package (version 3.1–3) was employed to obtain p-values for the fixed effects [32].

## Results

### Speech recognition

Individual and group mean speech understanding in noise is shown in Fig 1. Speech recognition in noise with the CI alone was completed with the contra ear unplugged. There was a statistically significant main effect of listening condition on speech recognition, $F(17,2) = 9.416$, $p < 0.001$. Post hoc analysis revealed significant differences in speech recognition between the CI+CROS ($M = 52.9$, $SD = 19.3$) and CI-alone ($M = 37.8$, $SD = 17.5$) listening conditions, and between the CI+HA ($M = 60.4$, $SD = 21.9$) and CI-alone listening conditions. The differences in speech recognition between CI+CROS and CI+HA were not statistically different (p >.05).

### Lateralization testing

Individual and group non-CI ear lateralization accuracy is shown in Fig 2. The mean (standard deviation) of these results are reported in percent correct. There was a statistically significant

## Speech Recognition in Noise

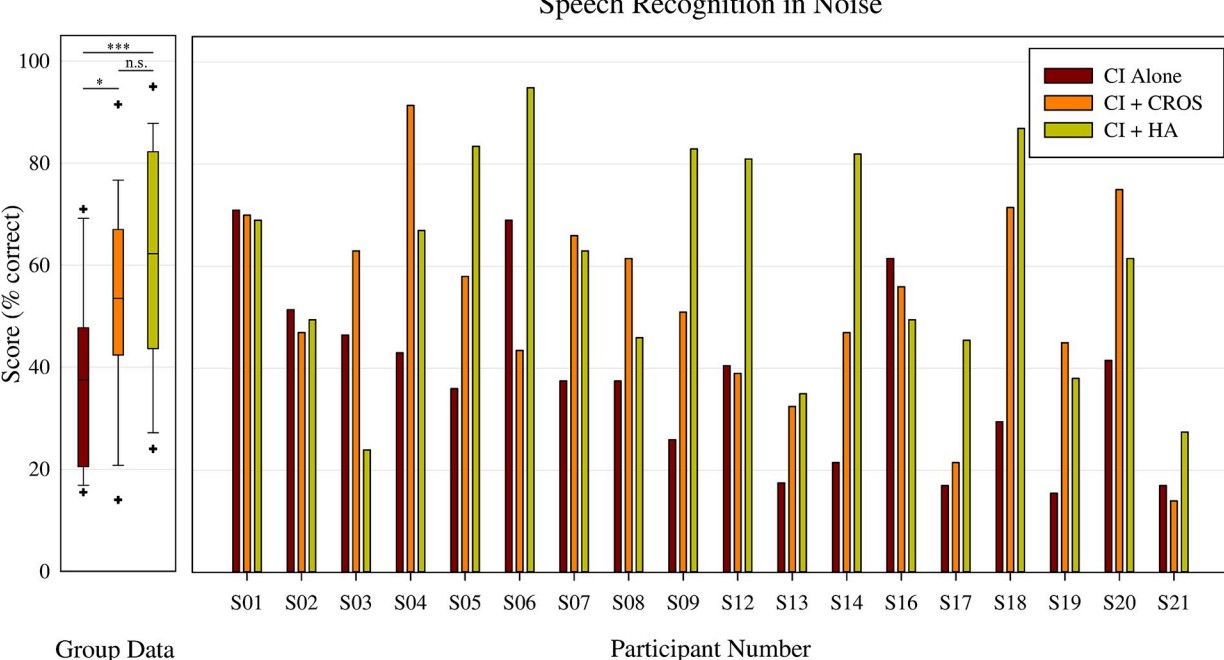

**Fig 1. Speech recognition in noise.** On the left, group data for CI alone (red), CI+CROS (orange), and CI+HA (yellow), where the whiskers represent the minimum and maximum scores for each listening condition, the '-' represents the median, and the '+' represent outliers. On the right, individual speech recognition scores for CI alone (red), CI+CROS (orange), and CI+HA (yellow). Participants 2, 3, and 20 expressed a preference for the CROS device at the end of the study.

## Lateralization

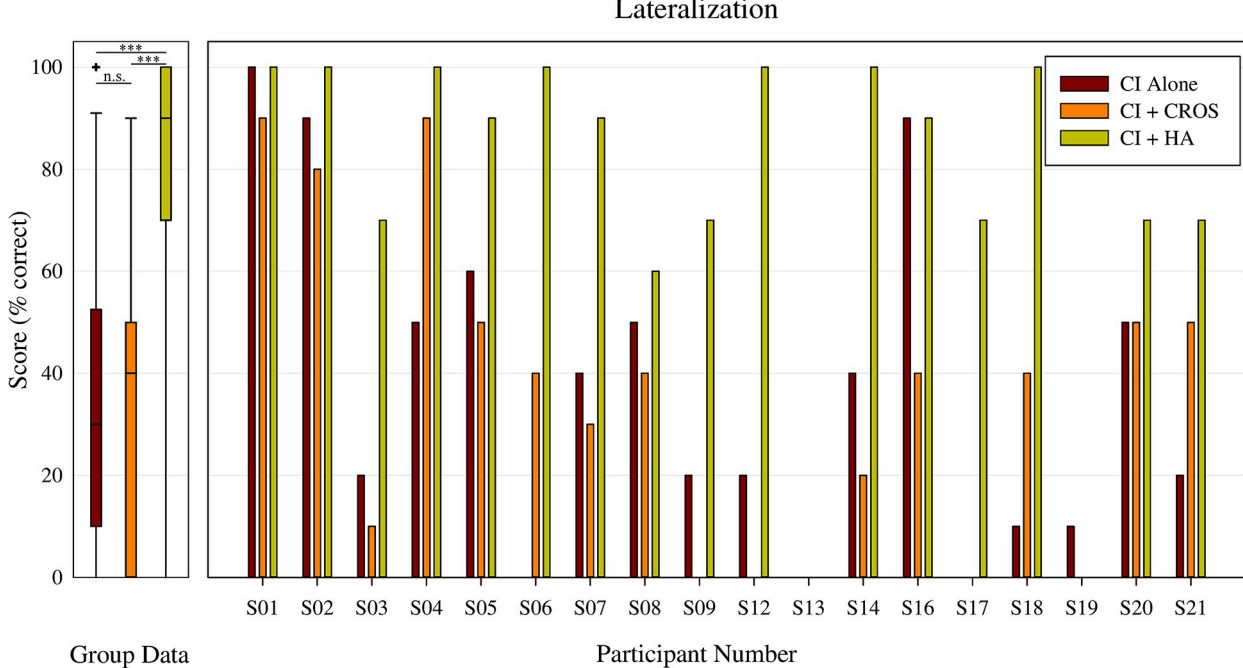

**Fig 2. Non-CI ear lateralization accuracy.** On the left, group data for CI alone (red), CI+CROS (orange), and CI+HA (yellow), where the whiskers represent the minimum and maximum scores for each listening condition, the horizontal bar represents the median, and the '+' represent outliers in our dataset. Median data are indicated by a a '-'. On the right, individual non-CI ear lateralization accuracy in percent correct for CI alone (red), CI+CROS (orange), and CI+HA (yellow). Data for CI alone testing was collected with the contralateral ear un-occluded. Participants 2, 3, and 20 expressed a preference for the CROS device at the end of the study.

main effect of listening condition on lateralization performance, $F(17,2) = 23.109$, p < 0.001. Post hoc analysis revealed significant differences in lateralization performance between the CI +HA and CI+CROS listening conditions, 76.7 (31.2) and 35 (30.5), respectively. A significant difference in lateralization performance between the CI+HA and CI-alone listening conditions was also observed, 76.7 (31.2) and 37.2 (32.0). The differences in lateralization performance between CI+CROS and CI-alone listening conditions were not statistically different (p >.05).

## Questionnaires

Responses to the baseline needs and chronic use questionnaires are summarized in Fig 3. The statements from the baseline needs and chronic use questionnaires were categorized into two domains: communication needs (panel A) and spatial awareness needs (panel B). On average, participants rated baseline spatial hearing needs 5.97 (1.39). For example, "I need to hear sounds from my non-CI side" had an average rating of 6.47 (.87). On average, participants rated baseline communication needs 6.59 (.80). For example, "I need to hear other people in a large group" had an average rating of 6.77 (.55). After chronic use of the CROS device,

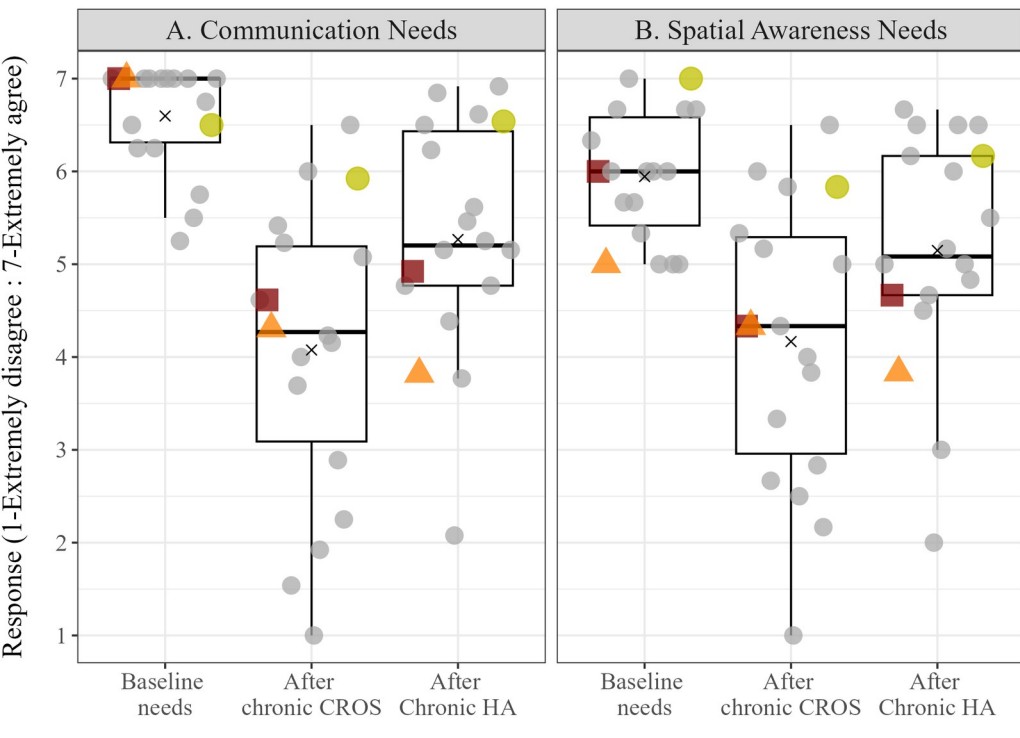

**Fig 3. Responses to the baseline needs and chronic use questionnaires.** Each boxplot represents the distribution of the average response from each of the 18 participants in the communication needs domain (panel A) and the spatial awareness needs domain (panel B) at baseline and after chronic CROS and HA use. The horizontal line within each boxplot represents the median, while the upper and lower boundaries of the boxplot indicate the interquartile range (IQR). The whiskers extend to the highest and lowest average within 1.5 times the IQR. Data points beyond this range are considered outliers. The "x" shape represents the average response collapsed across all participants for a particular domain. The colored symbols represent the responses from participant 2 (red square), participant 3 (orange triangle), and participant 20 (yellow circle) who opted to keep the CROS device at the end of the study, while the gray filled circles represent the individuals who chose to keep the HA at the end of the study.

participants were neutral in how the device met their communication needs ($M$ = 4.11, $SD$ = .35) and spatial awareness needs ($M$ = 4.16, $SD$ = .35). After chronic use of the HA, participants rated in the communication needs domain and the spatial awareness needs domain, 5.23 (.43) and 5.11 (.48), respectively; indicating that the HA device more closely met their needs in these domains.

## Effect of starting group

Linear mixed-effects models were employed to assess the impact of the order in which the devices (CROS or HA) were tested on the outcome measures. The analysis revealed no statistically significant effect of starting device on either speech or lateralization outcomes. For speech recognition, the estimated difference in means between starting with the HA and starting with the CROS device was -8.7125 (95% CI: [-27.4467, 10.0217], $p$ = 0.281), indicating no significant impact on speech performance. Similarly, in the lateralization task, the estimated difference was -17.25 (95% CI: [-47.5083, 13.0083], $p$ = 0.186), suggesting no significant effect on lateralization abilities.

The analysis also revealed no statistically significant effect of starting device on responses to the items in the spatial awareness or communication domains. For the self-reported items in the communication domain, the estimated mean difference between starting with the HA and starting with the CROS device was -0.4335, 95% CI: [-1.4843, 0.6173], $p$ = 0.408). For self-reported items in the spatial awareness domain, the estimated mean difference between starting with the HA and starting with the CROS device was -0.6396, 95% CI: [-1.7309, 0.4516], $p$ = 0.237).

## Exploratory analyses into audiometric slope of non-ci ear and benefit

As anticipated, speech understanding in noise was significantly higher with the addition of a contralateral device compared to CI-alone listening when target speech is presented at the non-implanted ear. While, as a group, scores with CI+HA and CI+CROS listening conditions were not statistically different, there were individual differences. To further analyze the individual differences, three groups were defined. The "HA Benefit (HAB) Group" was defined as individuals who received a greater than 10% improvement in speech understanding over their CI+CROS performance (HA benefit in noise—CROS benefit in noise > 10%). Eight participants met the HAB group criteria, averaging 29.9% (12.9) more benefit with the addition of a HA compared to the CROS. The "CROS Benefit (CB) Group" was defined as individuals who received a greater than 10% improvement in speech understanding over their CI+HA recondition in noise (CROS benefit in noise—HA benefit > 10%). Four participants met the CB group criteria, averaging 23.1% (11.6) more with the CROS when compared to the HA. Six individuals did not receive CROS or HA benefits of greater than 10%. These individuals averaged only a 3.75% (2.42) difference in CI+HA versus CI+CROS listening configurations and made up the "Equivalent Benefit (EB) group."

A greater objective of the study was to identify clinical guidelines for assessing which device may be of more benefit, a HA or a CROS, for unilateral CI recipients who have limited aidable hearing in the non-CI ear and are unwilling or unable to get a second CI. To this effect, differences in outcomes across the different test measures were investigated among the three groups defined above. Individual and average non-CI ear air conduction audiograms for each benefit group is shown in Fig 4A–4C. The percentage of aidable frequencies for each group was calculated as the total number of aidable thresholds divided by the total number of thresholds. A threshold was defined as aidable if it fell within the fitting range of Phonak's Naída Link M BTE, which is capable of fitting moderate to profound hearing loss of all audiometric

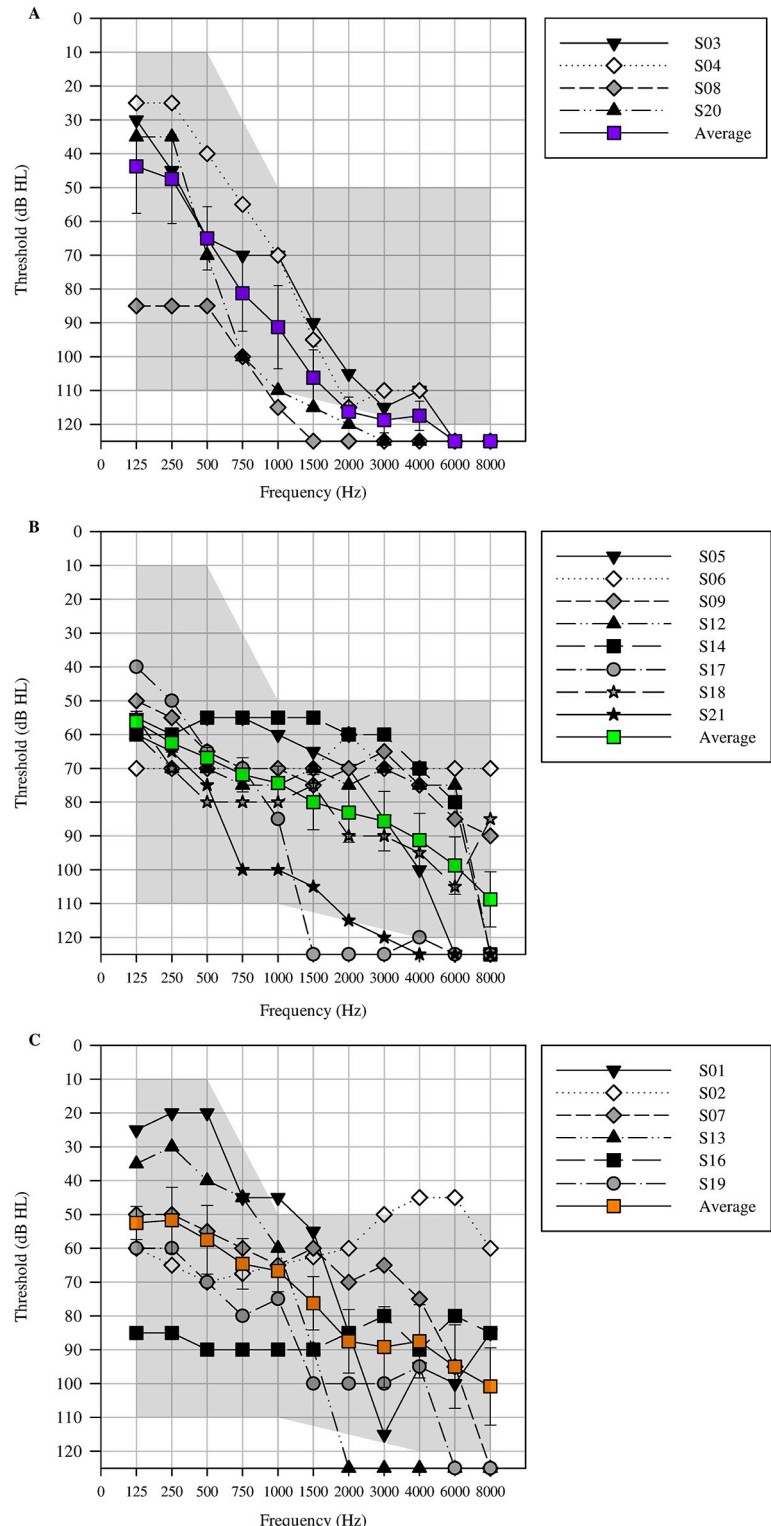

**Fig 4. Individual and mean unaided air conduction thresholds.** Individual and mean unaided air conduction thresholds for three groups (CROS benefit (panel A), HA benefit (panel B), and equivalent (panel C). Error bars indicate standard error. The shaded region indicates the fitting range of the Phonak Link M hearing aid.

configurations. Amidst the present study's participants, individuals with more steeply sloping hearing losses and fewer aidable frequencies (61% of frequencies were aidable) demonstrated greater CROS benefit. In contrast, individuals that received greater benefit from the HA, had flatter hearing losses and more aidable frequencies (85% of frequencies were aidable).

We investigated the slope of the hearing loss calculated as the hearing level difference from 250 to 750 Hz, calculated as follows:

$$[dB\,HL(f2) - dB\,HL(f1)]/\log2(frequency2/frequency1)$$

The average slope for the CB group, HAB group, and EB group, in dB/octave, was 21.3 (13.7), 5.92 (8.67), and 8.15 (5.50), respectively. In Fig 5, slope is plotted as a function of device benefit. Greater benefit tended to be associated with shallower audiometric slopes for all groups. While individuals that benefitted more from HA tended to have shallower sloping hearing losses from 250–750 Hz compared to HAB and EB groups, the smaller size and higher dispersion of slopes in the CB group make a complementary observation difficult and further analyses without a larger sample size is not possible without more data.

### End of study device preferences

A total of 18 participants were involved in the research investigation. After the study, 15 participants decided to retain the Hearing Aid (HA) they used throughout the study. Of the three individuals who opted to keep the CROS device, two were from the CB group, and one was from the EB group.

## Discussion

The present study assessed the value of a range of test measures in examining whether a hearing aid or CROS device provide greater benefit for a subset of unilateral CI users with limited aidable hearing in the non-implanted ear. Marked improvement in speech in noise and unimplanted ear lateralization accuracy were observed with the addition of a contralateral hearing aid. There were no significant differences in speech recognition between the hearing aid or CROS, and the findings of this study suggest that the order in which these devices were tested did not significantly influence a range of outcome measures However, the chronic device use questionnaires and the final device selection showed a clear preference for the hearing aid.

### Additional discussion points

In this study we saw an increase in speech understanding with the addition of a contralateral device. In the CROS listening condition, it is sensible to expect an improvement over the CI alone listening condition by mixing in the CROS signal, which has better signal-to-noise ratio due to its physical proximity to the target speech, which allows the user to overcome some of the deleterious effects of the head-shadow. These results are consistent with previous work with different test setups that have shown similar findings [17, 26]. Additionally, Dorman et al. [26] discusses how physical summation and adjustments applied to the T-Mic input, also mentioned in the methods above, could possibly explain why 14/17 participants across Dorman et al. [26] and Dwyer et al. [17] saw a benefit with the CROS over unilateral listening alone. However, in the current work speech was presented off axis and so it is more likely overcoming the head-shadow is where most of the benefit we observed could be attributed to.

The literature is less clear as to why we see the benefit by adding a hearing aid on the non-CI ear. Morera et al. [33] found a significant effect of binaural squelch in 12 bimodal adults after 6-months of CI use, however, they did not observe an effect of the head-show or

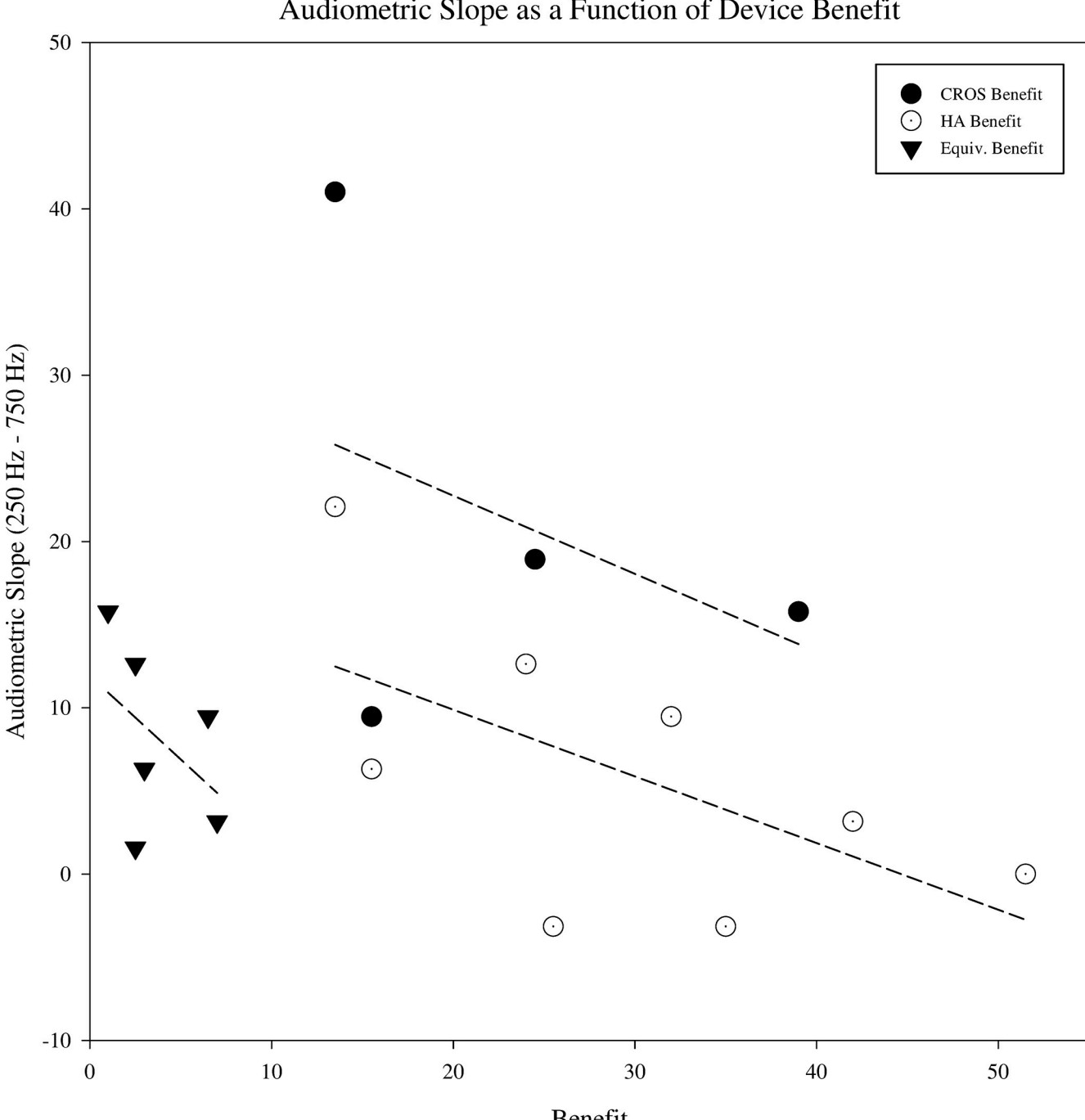

**Fig 5. Audiometric slope as a function of device benefit.** Audiometric slope (250 Hz -750 Hz) plotted as a function of benefit for each participant in the CROS benefit group (filled black circles), the HA benefit group (open circles), and the equivalent benefit group (filled triangles). The dashed line represents a simple linear regression for each group.

summation effect. Contrary to the work of Morera et al. [33], a more recent meta-analyses from Schafer et al. [34] found just the opposite; no effect of binaural squelch, but medium effect sizes for head-shadow and small-medium effects of binaural summation were observed.

Taken together, the benefit from the addition of a contralateral device is a likely result of providing these listeners with access to certain binaural advantages.

Some participants benefited more from a CROS device than a hearing aid; others, the opposite. Others showed an equivalent level of benefit with both devices. A small subset of individuals showed that adding a contralateral device could be detrimental (i.e., worse than listening in the CI alone condition). It would be beneficial for clinicians to know which device to fit their patients with, if any.

Our exploratory analyses indicates that audiometric slope could explain why some individuals benefit with one device over the other. In the context of the lateralization task, it is noteworthy to highlight the performance of participants S14 and S16, who had the largest performance declines compared to CI alone following the incorporation of the CROS device. This observation is notable, given that both participants have flat audiometric configurations, as shown in Fig 4.

In the speech recognition task, S3's performance in the CI+HA condition was worse compared to their unilateral CI performance. Notable is this individual's steeply sloping hearing loss in the non-implanted ear. Contrary to participant S3's hearing loss configuration and detriment with the addition of a HA, S6's has a flat hearing loss and performance that deteriorated upon the introduction of the CROS device.

The current work comes on the heels of other work in this area that has recently investigated the level of residual hearing and the impact on CROS benefits [35]. Stronks et al. [35] showed that when the speech signal was presented to the CROS ear and noise presented to the CI ear, SRT improved by 6.4 dB on average, compared to CI alone. However, the amount of residual hearing in the non-implanted ear was significantly correlated with CROS benefit. Taken together, audiometric thresholds in individuals with limited aidable hearing in the non-CI ear play an important role in which device could be prescribed.

## Subjective questionnaires and end of study device preferences

While speech intelligibility results are a driving factor in making a clinical recommendation in selecting a device, a growing body of literature indicates the importance of self-perceived benefit when evaluating an intervention. Gifford et al. [16] investigated participants' thoughts on whether they believed they would benefit from a second CI. In the absence of objective data, they found that simply asking a bimodal patient if they thought they needed a second CI was sensitive enough to identify likely bilateral CI candidates. Therefore, the current study included baseline and chronic use subjective questionnaires to assist with evidence-based, clinic-based outcome measures to help guide decisions in clinical practice. This information was particularly helpful for this subset population of unilateral CI users where the difference in speech recognition performance in noise was not statistically different between listening conditions, yet there was a clear device preference.

Subjective reports were further examined as a function of each benefit group. Overall, the CB group rated items in both the communication ($M = 4.9$, $SD = 1.06$) and spatial hearing domains ($M = 5.13$, $SD = 1.08$) lower than how they rated these domains after chronic HA use (communication domain: $M = 5.35$, $SD = 1.41$; spatial hearing domain: $M = 5.33$, $SD = 1.37$). However, the two individuals in the CB group that provided higher ratings with CROS, relative to the two that showed a preference for the HA, interestingly chose to keep the CROS device at the end of the study. This highlights the importance of investigating self-perceived benefit in clinical decision making.

In contrast to the CB group, we saw agreement between the subjective questionnaires and clinic-based outcome measures within the HAB group. These participants rated items in the

communication domain higher after the chronic HA take home period ($M = 4.96$. $SD = 1.78$) than they rated these same items after the CROS take home period ($M = 3.57 = SD = 1.96$). The HAB group also rated items in the spatial hearing domain greater after the HA trial, 4.88 (1.77) vs. 3.58 (1.92). Subsequently, all eight participants in the HAB group decided to keep the hearing aid after the study, emphasizing the value of spatial awareness and speech recognition with bilateral input in everyday situations.

Of the six participants in the EB group, only one chose to keep the CROS device at the conclusion of the study. Those who decided to keep the HA, rated the spatial needs domain items ($M = 5.68$, $SD = 1.17$) and communication domain items ($M = 5.88$, $SD = 1.22$) greater after the HA take home period than the spatial items ($M = 4.58$, $SD = 1.84$) and communication domain items ($M = 4.34$, $SD = 1.76$) after the CROS take home period. The participant in the EB group who chose to keep the CROS was a bimodal listener prior to study participation and rated spatial hearing and communication items for both devices similarly. This individual rated items in the both the spatial hearing domain (4.33 for CROS v. 4.67 for HA) and in the communication domain (4.62 for CROS v. 4.92 for HA) after chronic trial with both devices.

While all but one individual in the EB group was a bimodal user prior to study enrollment indicating a possible bias for the HA, the cues conveyed through acoustic hearing are more likely to explain why 5/6 of the participants in this group chose to keep the HA at the conclusion of the study. Previous work has shown that some, albeit limited, binaural processing with bimodal listening may allow bimodal listeners to take advantage of binaural squelch and summation effects, which may benefit them in complex listening situations, such as noise [33, 36]. Additionally, bimodal listeners may have access to ILD cues that they did not have access to while listening in the CI+CROS condition, depending on the amount of aidable hearing, the frequency content of a signal, and the spectral overlap between the HA and CI [37]. This may have contributed to participants' increased ratings in the spatial hearing and communication domains and explain the preference for the HA over the CROS at the end of the study.

### Limitations and future directions

The authors recognize several limitations that may affect the validity and reliability of the study findings. Nearly all (15/18) participants in this study were current bimodal listeners, indicating a possible bias for the HA condition. Future studies may consider enrolling subjects without any contralateral device use experience. We also did not employ data logging to monitor device use compliance during the take-home periods. Recent work has shown the influence of device use duration on outcomes [38–40], and further investigation of the effects of device use should be documented in all clinical studies in the future. It is also possible that the addition of tasks that assess cochlear dead regions, such as the TEN (Threshold Equalizing Noise) test [41] and an additional measure of spectral resolution, could further inform the clinician as to which device to choose for their patient. Future work in this area should consider adding one (or both) of these measures.

An important consideration in our study was the selection of a suitable questionnaire to assess the communication needs and the extent to which the evaluated hearing devices met those needs. While established and validated questionnaires, such as the SSQ [42], are commonly employed in research endeavors of this nature, we chose to develop a novel questionnaire tailored to the specifics of our research objectives.

The decision not to adopt an existing validated questionnaire is acknowledged as a limitation of our study, as validated questionnaires provide a standardized framework for data collection and facilitate inter-study comparisons. We recognize that future studies in this domain may benefit from considering a dual-method approach, incorporating both established

validated instruments and context-specific questionnaires to strike a balance between standardization and the uniqueness of research objectives.

Furthermore, we recognize that elements of this study relied on self-reported data, which may be subject to biases and inaccuracies. While efforts were made to ensure that the questionnaires were developed with expert input and tested for clarity, participants may still have different interpretations of the questions. Additionally, the consistency of questionnaire administration may not account for variations in individual interpretation or understanding of the questions.

Additionally, data collection was completed just prior to the launch of Marvel CI, however, we do not believe that using previous generation technology necessarily detracts from the study findings as the technology used was representative of rehabilitative options during this time, and the findings may still be relevant to current discussions and debates. Lastly, the authors recognize that this is a small study and that these results may not be generalized to the population as a whole; thus, larger-scale investigations are warranted.

## Conclusions

The objective of this study was to explore potential clinical guidelines for assessing which device may be of more benefit, a HA or a CROS, for unilateral CI recipients who have limited bimodal benefit and are unwilling or unable to get a second CI. Participants in this study saw a marked improvement with the addition of a contralateral device. Both device configurations significantly improved speech in noise for IEEE sentence recognition. However, 15 out of 18 participants chose to keep and preferred the HA at the conclusion of the study, despite only eight participants having received greater benefit with the HA for speech in noise. The chronic device use questionnaires explain this contrast, which showed a general preference for the hearing aid for improved spatial awareness and communication in both individuals who scored higher with the CROS device and in others that benefitted equally in speech recognition tasks with both devices. This suggests that overall device preference was more dependent upon perceptive benefit for spatial awareness and communication needs and highlights the need for additional investigation and the delicate decision-making process involved in clinically advising one device or another for overall communication benefit.

## Supporting information

**S1 Dataset. Individual data for all participants.**
(XLSX)

**S1 File.**
(ZIP)

## Author Contributions

**Conceptualization:** Sarah Lively, Smita Agrawal.

**Data curation:** Sarah Lively, Robert T. Dwyer.

**Formal analysis:** Robert T. Dwyer.

**Investigation:** Sarah Lively, Matthew Stewart, Laura Strobel, Paula Marcinkevich.

**Methodology:** Sarah Lively, Smita Agrawal.

**Project administration:** Julia Croce.

**Resources:** Julia Croce.

**Software:** Chris Hetlinger.

**Supervision:** Smita Agrawal.

**Visualization:** Sarah Lively, Smita Agrawal, Robert T. Dwyer.

**Writing – original draft:** Sarah Lively, Smita Agrawal, Matthew Stewart, Robert T. Dwyer, Laura Strobel, Paula Marcinkevich.

**Writing – review & editing:** Sarah Lively, Smita Agrawal, Matthew Stewart, Robert T. Dwyer, Laura Strobel, Paula Marcinkevich.

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
