## [Decision Letter · Decision Letter 0]

14 Aug 2023

PONE-D-23-13205CROS or Hearing Aid? Selecting the ideal solution for unilateral CI patients with limited aidable hearing in the contralateral earPLOS ONE

Dear Dr. Lively,

Thank you for submitting your manuscript to PLOS ONE. After careful consideration, we feel that it has merit but does not fully meet PLOS ONE’s publication criteria as it currently stands. Therefore, we invite you to submit a revised version of the manuscript that addresses the points raised during the review process.

We look forward to receiving your revised manuscript.

Kind regards,

Paul Hinckley Delano, Ph.D.

Academic Editor

PLOS ONE

Journal Requirements:

"I have read the journal's policy and the authors of this manuscript have the following competing interests: RD, CH, and SA are employees of Advanced Bionics. The remaining authors declare no conflicts of interest. This does not alter the authors’ adherence to all the PLOS ONE policies on sharing data and materials. There are no patents, products in development or marketed products to declare."

6. Please upload a new copy of all Figures as the detail is not clear. Please follow the link for more information: " ext-link-type="uri" xlink:type="simple">https://blogs.plos.org/plos/2019/06/looking-good-tips-for-creating-your-plos-figures-graphics/"
https://blogs.plos.org/plos/2019/06/looking-good-tips-for-creating-your-plos-figures-graphics/

Additional Editor Comments:

Add colors to figures 1 and 2

Reviewers' comments:

Reviewer's Responses to Questions

**Comments to the Author**

1. Is the manuscript technically sound, and do the data support the conclusions?

Reviewer #1: Yes

Reviewer #2: Yes

2. Has the statistical analysis been performed appropriately and rigorously? 

Reviewer #1: Yes

Reviewer #2: Yes

3. Have the authors made all data underlying the findings in their manuscript fully available?

Reviewer #1: Yes

Reviewer #2: Yes

4. Is the manuscript presented in an intelligible fashion and written in standard English?

Reviewer #1: Yes

Reviewer #2: Yes

5. Review Comments to the Author

Reviewer #1: Excellent article regarding the need to evaluate whether CROS or hearing aids are best for patients with unilateral CI and hearing loss in the contralateral ear. It is a debatable question as it depends on the patients perception but the authors aimed to assess various aspects including a questionnaire, and the inclusion criteria were adequate.

Reviewer #2: Attached

The manuscript by Lively et al. is a well written report about the benefits of bimodal listening, aiding the contralateral ear in unilateral CI patients. The authors show significant improvement in speech in noise and lateralization in most patients when a haring aid or CROS device were implemented in the non-implanted ear. They also discussed the limitation of the study regarding small sample size and previous use of hearing aid devices.

Overall, methodology is clear and could be replicated, the data and figures are well laid-out but could benefit from showing plots including self-reported information. Statistics are well reported in the manuscript, but the analysis of the slopes is not appropriate to the data considering the small sample size.

Comments:

• Participants were randomized to either HA or CROS group for phase 1 and then fitted with the other device on phase 2. Was there any effect of the starting group on speech/localization or self-reported measures?

• Although questionnaires were developed with the assistance of experts, could you have used other questionnaires used by other research/clinical groups? This helps when comparing the outcomes of studies like this one, improving the impact of the evidence.

• According with other studies and your results, investigating the slope of the hearing loss could give more information to help the decision of which device to select in order to provide more benefit to the patient. However, the statement of grater benefit = shallower slopes in all groups is not clear as showed in Figure 4, whit only 4 subjects in the CB group and high dispersion. The statement is clearer in the HA group. More important could be reporting the slope (using correlation test) of the fitted line in each condition than using ANOVA for comparing the group, which could be useful with a bigger sample size.

• Please report the final device selection in the results section.

• I do agree with the authors and evidence that the self-reported measures are important when selecting an intervention like the ones reported here. I suggest showing the association of data from questionnaires, behavioral measures, and final device selection. This could be helpful to understand the complexity of the decision-making process, specially because most of the participants of this study chose HA over CROS device for the un-implanted ear even if the later one showed to be more beneficial for some of them in speech in noise and lateralization of sounds.

• The results of this work also show patients with no benefit (ie. better performance with CI-only) in comparison with HA and/or CROS devices. This issue should also be discussed in here (Ln 330-331). Only reporting the benefit with HA or CROS or equivalent benefit is bias.

Minor comments:

• ID of participants on table 1 must match the ID number used in the plots. For example, in Fig 1 and 2 the authors used subject ID’s from 1 to 18, but in Table 1 and Fig 3, they used S1 to S21.

• Figures and statistics: it is necessary to include statistically significant and non-significant differences in the figures (eg. Fig 1 and 2).

• Figure 2: “*” representing median data for CI+CROS condition is missed. Please check and include the symbol.

• Figure 3: Please show (eg. Shaded area) or describe the fitting range for the definition of “aidable threshold”. Also, we suggest including this information in the text, line 262-263.

• Line 350: replace , with .

6. PLOS authors have the option to publish the peer review history of their article (what does this mean?). If published, this will include your full peer review and any attached files.

Reviewer #1: No

Reviewer #2: No

---

## [Author Response · Author response to Decision Letter 0]

27 Sep 2023

All reviewer and editor comments are in the file named Response to reviewers attached to this submission.

---

## [Editor Report · Decision Letter 1]

20 Oct 2023

CROS or Hearing Aid? Selecting the ideal solution for unilateral CI patients with limited aidable hearing in the contralateral ear

PONE-D-23-13205R1

Dear Dr. Lively,

We’re pleased to inform you that your manuscript has been judged scientifically suitable for publication and will be formally accepted for publication once it meets all outstanding technical requirements.

Kind regards,

Paul Hinckley Delano, Ph.D.

Academic Editor

PLOS ONE
---

## [Editor Report · Acceptance letter]

27 Oct 2023

PONE-D-23-13205R1 

CROS or Hearing Aid? Selecting the ideal solution for unilateral CI patients with limited aidable hearing in the contralateral ear 

Dear Dr. Lively:

I'm pleased to inform you that your manuscript has been deemed suitable for publication in PLOS ONE. Congratulations! Your manuscript is now with our production department. 

Kind regards, 

on behalf of

Dr. Paul Hinckley Delano 

Academic Editor

PLOS ONE